# Maternal consumption of yoghurt activating the aryl hydrocarbon receptor increases group 3 innate lymphoid cells in murine offspring

Grégory Pimentel,[1] Thomas Roder,[2] Cornelia Bär,[1] Sandro Christensen,[3,4] Zahra Sattari,[1,4] Cristina Kalbermatter,[3,4] Ueli von Ah,[1] Christelle A. M. Robert,[5] Pierre Mateo,[5] Rémy Bruggmann,[2] Stephanie C. Ganal-Vonarburg,[3,4] Guy Vergères[1]

**ABSTRACT** Indole derivatives are microbial metabolites of the tryptophan pathway involved in gut immune homeostasis. They bind to the aryl hydrocarbon receptor (AhR), thereby modulating development of intestinal group 3 innate lymphoid cells (ILC3) and subsequent interleukin-22 production. In mice, indole derivatives of the maternal microbiota can reach the milk and drive early postnatal ILC3 development. Apart from the gut microbiota, lactic acid bacteria (LAB) also produce indole compounds during milk fermentation. Using germ-free mice, the aim of our study was to test if maternal intake of a dairy product enriched in AhR-activating indoles produced by fermentation could boost maturation of the intestinal innate immune system in the offspring. A set of 631 LAB strains were genetically screened for their potential to produce indole compounds. Among these, 125 strains were tested in combination with standard strains to produce yoghurts that were screened for their ability to activate AhR *in vitro* using the HepG2–AhR–Luc cell line. The most active yoghurt and a control yoghurt were formulated as pellets and fed to germ-free dams during pregnancy and lactation. Analysis of the offspring on postnatal day 14 using flow cytometry revealed an increase in the frequency of small intestinal lamina propria NKp46 +ILC3 s in the pups born to dams that had consumed the purified diet containing an AhR-active yoghurt (AhrY-diet) compared to control yoghurt (ConY-diet). Selection of LABs based on their ability to produce a fermented dairy able to activate AhR appears to be an effective approach to produce a yoghurt with immunomodulatory properties.

**IMPORTANCE** Key progresses in the sequencing and functional annotation of microbial organisms have revolutionized research in the fields of human metabolism and food biotechnology. In particular, the gut microbiome is now recognized as an important mediator of the impact of nutrition on human metabolism. Annotated genomes of a large number of bacteria are now available worldwide, which selectively transform food through fermentation to produce specific bioactive compounds with the potential to modulate human health. A previous research has demonstrated that the maternal microbiota shapes the neonatal immune system. Similarly, this report shows that lactic acid bacteria can be selected to produce fermented food that can also modulate postnatal intestinal immunity.

**KEYWORDS** aryl hydrocarbon receptor, lactic acid bacteria, newborn immunity

F ermented foods constitute 20% to 40% of the current world food supply (1, 2). Although residues of grapes and fermentation markers dating from 4300 BC were discovered in a jar in Greece (3), genetic evidence suggests that adaptation of hominids to naturally fermented foods, in particular ethanol in ripened fruits, may have taken

**Peer Reviewer** Manoj Gurung, USDA-ARS Arkansas Children's Nutrition Center, Little Rock, Arkansas, USA

Address correspondence to Guy Vergères, guy.vergeres@agroscope.admin.ch.

Stephanie C. Ganal-Vonarburg and Guy Vergères contributed equally to this article.

The authors declare no conflict of interest.

See the funding table on p. 14.

place 10 million years ago (4). The selective advantages of food fermentation include food preservation (5), modification of the organoleptic character of the food matrix (6), and health benefits (7, 8). Fermentation transforms food constituents (e.g., breakdown of proteins to produce bioactive peptides or increase the bioavailability of free amino acids), synthesizes bioactive and nutritive compounds (e.g., B vitamins), and promotes a healthy gut microbiome by delivering commensal microbes and probiotics to the gastrointestinal tract (9). These properties of fermented foods have been associated with potential benefits for a broad range of organ systems including the digestive, cardiovascular, and nervous systems (10). The field of microbiology has been revolutionized during the last decade based on the technological breakthroughs in DNA sequencing and biocomputing. These technologies have contributed to validate the importance of the gut microbiota in human health (11) as well as the key role that nutrition has in modulating the structure and dynamics of the gut microbiota (12) and the immune system (13, 14).

Many metabolites that play a role in human health are derived from metabolization of dietary components by the gut microbiota. These include the transformation of choline to trimethylamine (TMA) and trimethylamine-N-oxide (TMAO), the transformation of primary bile salts to secondary and tertiary bile salts, the digestion of indigestible dietary fiber to produce short-chain fatty acids (SCFA), as well as the production of immuno-modulatory indoles from tryptophan (15). These compounds may also be metabolized during the fermentation of food by microorganisms, suggesting that the consumption of fermented foods may modulate the delivery of bioactive nutrients otherwise produced by the human gut microbiome. This has been shown for methylamines (16), bile acids, and indoles (17). In line with these findings, we (18) recently analyzed in silico the pan-genome of a collection of over 600 genomically annotated lactic acid bacteria (LAB) and found that a subset of 24 of these strains, each from a different species, covered 89% of the enzymatic reactions of the human gut microbiome. Targeted selection of bacteria for food fermentation therefore has significant potential for the production and delivery of bioactive substances to the human organism.

Indoles are tryptophan metabolites that represent an interesting group of bioactive molecules to target in biotechnology. Indoles potentially contribute to intestinal health and immune regulation through activation of the aryl hydrocarbon receptor (AhR) and the pregnane X receptor (PXR) (19) as well as to an array of additional properties associated with diabetes mellitus or vascular regulation (20, 21). Hence, one strategy to modulate immune responses is the use of nutritional AhR ligands (22) produced as bioactive indoles through food fermentation (17, 23). Murine studies have shown that dietary AhR ligands affect the formation of intestinal lymphoid follicles by expansion of small intestinal AhR-expressing type 3 innate lymphoid cells (ILC3) (24) and maintenance of intraepithelial lymphocytes (25). ILC3s produce the cytokine interleukin-22 (IL-22) acting on IL-22-receptor-expressing intestinal epithelial cells, which, in turn, respond with the production of anti-microbial peptides (26, 27). ILC3s have also been shown to prevent translocation of intestinal microbes to systemic sites (28) and to the defense against the enteric pathogen *Citrobacter rodentium* (24, 29). Interestingly, we could show with a model of reversible colonization of pregnant germ-free mice that indole metabolites produced by the maternal microbiota can reach the offspring via breast milk and can drive early postnatal expansion of a subset of intestinal ILC3s in the offspring (30).

In this report, we addressed whether LAB could be selected based on their genomic content to produce indole derivatives, be used to produce a yoghurt with increased levels of such compounds, and that would be able to activate the AhR. By feeding pregnant and lactating germ-free mice with purified diets containing an AhR-active yoghurt (AhrY-diet), we also assessed if this strategy can be used for a nutritional modulation of the innate immune system development of the offspring.

## RESULTS

### Selection of strains for production of indole-rich yoghurt

To select lactic acid bacteria (LAB) that potentially synthesize AhR-activating ligands, the genomes of 663 strains from our strain collection were screened for potential production of indole and indole derivatives as described in the Material and Methods section. Based on the *in silico* screening, 125 strains belonging to seven different genera were chosen (*Acidipropionibacterium*, *Lacticaseibacillus*, *Lactococcus*, *Leuconostoc*, *Pediococcus*, *Propionibacterium*, and *Streptococcus*). One hundred and twenty-five test yoghurts were produced by fermenting lactose-free cow's milk with each of the selected strains in combination with a standard yoghurt starter culture (composed of one strain of *Lactobacillus delbrueckii* subsp. *bulgaricus* and two strains of *Streptococcus salivarius* subsp. *thermophilus*). A control yoghurt was produced by fermenting milk with the yoghurt starter culture only. All yoghurts were screened *in vitro* for their ability to activate AhR using the HepG2–AhR–Luc cell line in which the luciferase gene is controlled by an AhR-response element. Luminescence signals (Fig. S1) showed that 19 yoghurts induced significant AhR activation compared to the control yoghurt (Wilcoxon signed-rank test $P < 0.05$). Based on these results, a yoghurt with expected high AhR activity (AhR yoghurt) was designed. The AhR yoghurt consisted of milk fermented with five bacterial strains from four species, i.e., the three starter strains associated with the strain that gave the highest AhR activation signal (*Lacticaseibacillus paracasei*) and another strain (*Lactococcus lactis* subsp. *cremoris*) chosen to increase species diversity in the AhR yoghurt. The 16 yoghurts with the highest AhR activation in the first round and the control yoghurt were then compared to the AhR yoghurt, the latter showing a two- to threefold increase in AhR activation compared to the control yoghurt (Wilcoxon signed-rank test $P = 0.024$, Fig. 1).

### Analysis of indole derivatives in the yoghurts

As a further read-out to characterize indole metabolites and derivatives in the yoghurts, we performed a targeted analysis using ultra-high performance liquid chromatography–mass spectrometry (UHPLC-MS) of a panel of 38 indole derivatives in the AhR- and control yoghurt, as well as in non-fermented milk. Of the 38 compounds belonging to the tryptophan pathway, 13 were detected in the test products (Table S1), with eight of these showing a significant difference between the AhR- and control yoghurts ($P < 0.05$): tryptophan, tryptophol, kynurenic acid, 5-hydroxy-L-tryptophan, indole-3-pyruvic acid, nicotinamide, anthranilic acid, and nicotinic or picolinic acid (the two acids could not be differentiated based on their masses or retention time) (Fig. 2A through H). Anthranilic acid was the only metabolite showing higher levels in the control yoghurt. These findings showed that the increased AhR activity of the AhR yoghurt was paralleled by increases in the concentrations of indole derivatives.

### Production and composition of the purified diets containing yoghurt for mice

To test the bioactivity of the AhR yoghurt on the immune system *in vivo*, we incorporated this dairy product into murine diet. The AhR yoghurt was lyophilized and incorporated at 40% (wt/wt) into an open standard purified diet (AhrY-diet) at Research Diets, Inc. (USA). A control diet (ConY-diet) was produced using the control yoghurt (macro- and micronutrient composition of the diets in Table S2). A major quality control following production was the sterility of these diets as yoghurt displays a high bacterial load. Absence of any living microbe was pre-requisite to our murine studies to be able to study the direct impact of the AhR ligands on the host immune system without the confounding factor of live microbes that can directly be sensed by and influence the intestinal immune system. The diets were sterilized by irradiation and tested for sterility by microbiological culture-dependent and -independent methods and by *in vivo* testing in germ-free mice, which were fed with the diets for 4 weeks and maintained their sterile status.

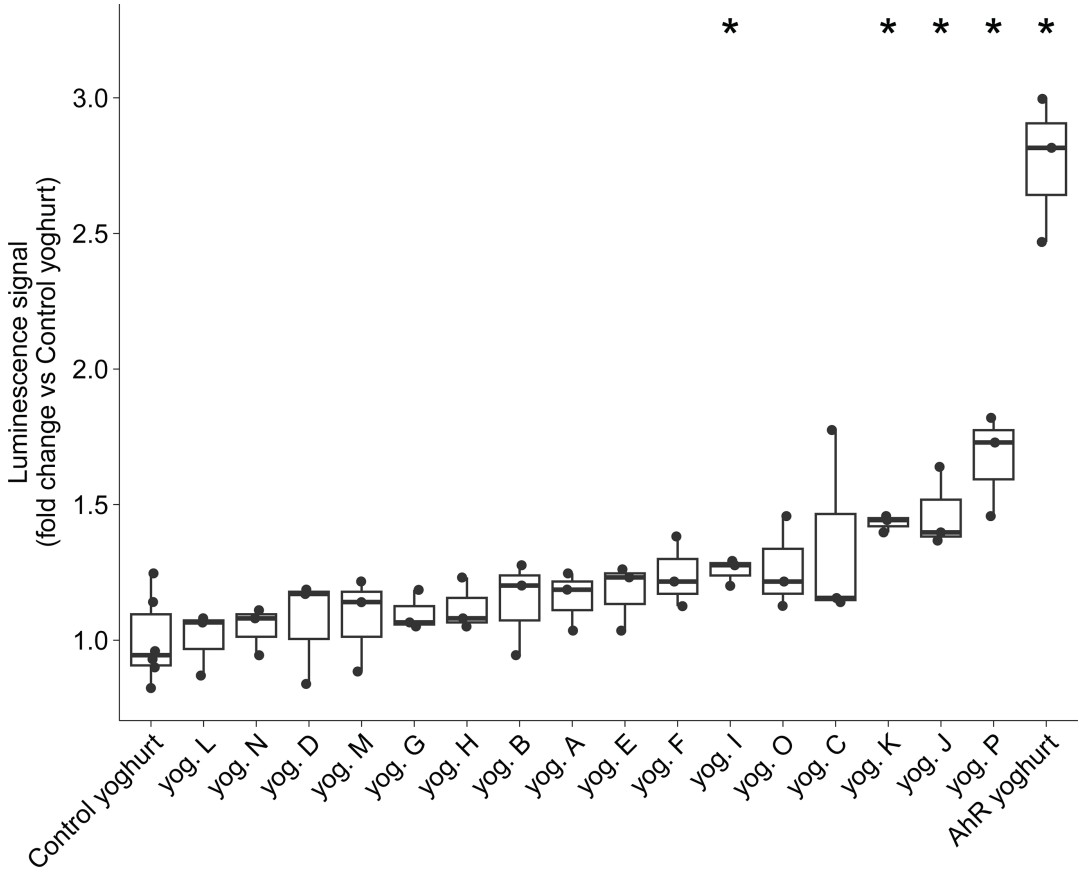

**FIG 1** *In vitro* AhR activation assay of 16 test yoghurts compared to the conventional control yoghurt. A total of 125 bacterial strains were tested to produce yoghurts that were then screened for their ability to activate AhR *in vitro* using the HepG2–AhR–Luc cell line. Two strains were then selected to produce the AhR yoghurt. The figure presents the ability of the top 16 strains and the AhR yoghurt to activate AhR *in vitro* compared to the control yoghurt (Wilcoxon test $P < 0.05$ as a threshold).

Among the eight indole derivatives differing in the yoghurts, tryptophol and nicotinic or picolinic acid were significantly higher in the AhrY-diet compared to the ConY-diet (Fig. 2J and K). Significant differences in tryptophan were not measured (Fig. 2I), and the remaining five metabolites could not be detected in the murine diets. These findings confirmed the selection of the AhR- and control yoghurts for formulation of the murine diets for functional tests.

## Mice born to germ-free dams consuming the AhrY-diet harbor increased numbers of ILC3s in the intestine

Germ-free pregnant mice were switched to either of the two different yoghurt-containing diets at day 7 post conception and kept on this diet until postnatal day 10. Between postnatal days 10 and 14, dams were fed a conventional chow diet, to prevent the pups from eating the yoghurt-containing diet themselves, which is possible from postnatal day 12 onward. On postnatal day 14, the immune compartment in the small intestinal lamina propria of the offspring was analyzed by flow cytometry (Fig. 3A). Germ-free mice were used as a model to exclude a secondary impact of an altered maternal or infant microbiota after consumption of either diet (AhrY-diet, ConY-diet). The effect of AhR ligands, which were fed to the pregnant dam and transferred to the pups via breast milk, on the frequency of ILC3 in the offspring small intestine was previously demonstrated using germ-free mice and timed colonization or application of AhR ligands during pregnancy (30). As previously, feeding AhR ligands to the dams, in our case via consumption of the AhrY-diet compared to the ConY-diet, was associated with an increase in the

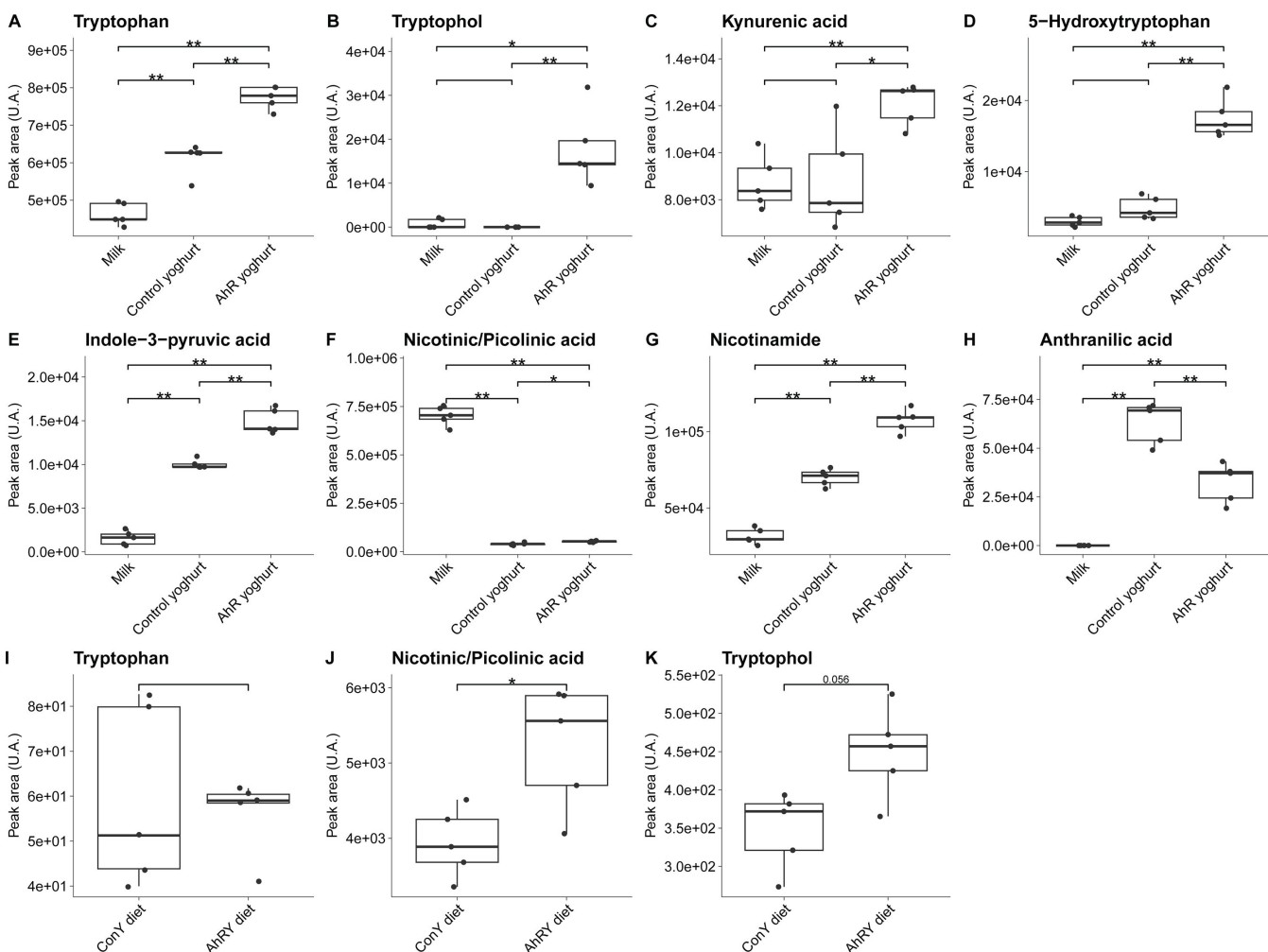

**FIG 2** Indole derivatives discriminating the test and control products. Targeted UHPLC-MS metabolomics was used to evaluate the relative contents of 38 indole derivatives and compounds belonging to the tryptophan pathway. (A–H) Eight compounds showed significant differences between the AhR yoghurt and the control yoghurt. Their relative concentration is also shown in the milk used to prepare the yoghurts. (I) Tryptophan did not show significant differences between the AhrY-diet and the ConY-diet. (J, K) Two compounds showed significant differences between the test diet (AhrY-diet) and the control diet (ConY-diet). Statistical significance is based on the Wilcoxon test with $P < 0.05$.

frequency of NKp46$^+$ILC3s in the small intestinal lamina propria of the offspring at postnatal day 14 (Fig. 3B and C). This effect was specific to ILC3s, as ILC2 and T-cell populations were not altered in frequency (Fig. S2) between the experimental groups.

## The composition of the milk and serum of germ-free dams fed the AhrY-diet differs from that of dams fed the ConY-diet

On postnatal day 10, serum and breast milk were collected, and the samples were analyzed using mass spectrometry for indole derivatives and their untargeted metabolome. No significant differences in the relative concentration of the indole derivatives were observed between the experimental groups. However, the untargeted multivariate analysis of the metabolomes (Fig. 4; Fig. S3) could clearly differentiate serum (A) and breast milk (B) of dams fed the AhrY-diet from dams fed the ConY-diet. The untargeted metabolome analysis could also differentiate the tested foods, i.e., the AhR yoghurt from the control yoghurt (C) as well as the AhrY-diet from the ConY-diet (D).

Pathway analysis of mice milk metabolomes using the mummichog tool revealed four pathways differentiating murine milk from AhrY-diet fed dams from control dams: linoleate metabolism ($P$ value = $8.71 \times 10^{-5}$), prostaglandin formation from arachidonate

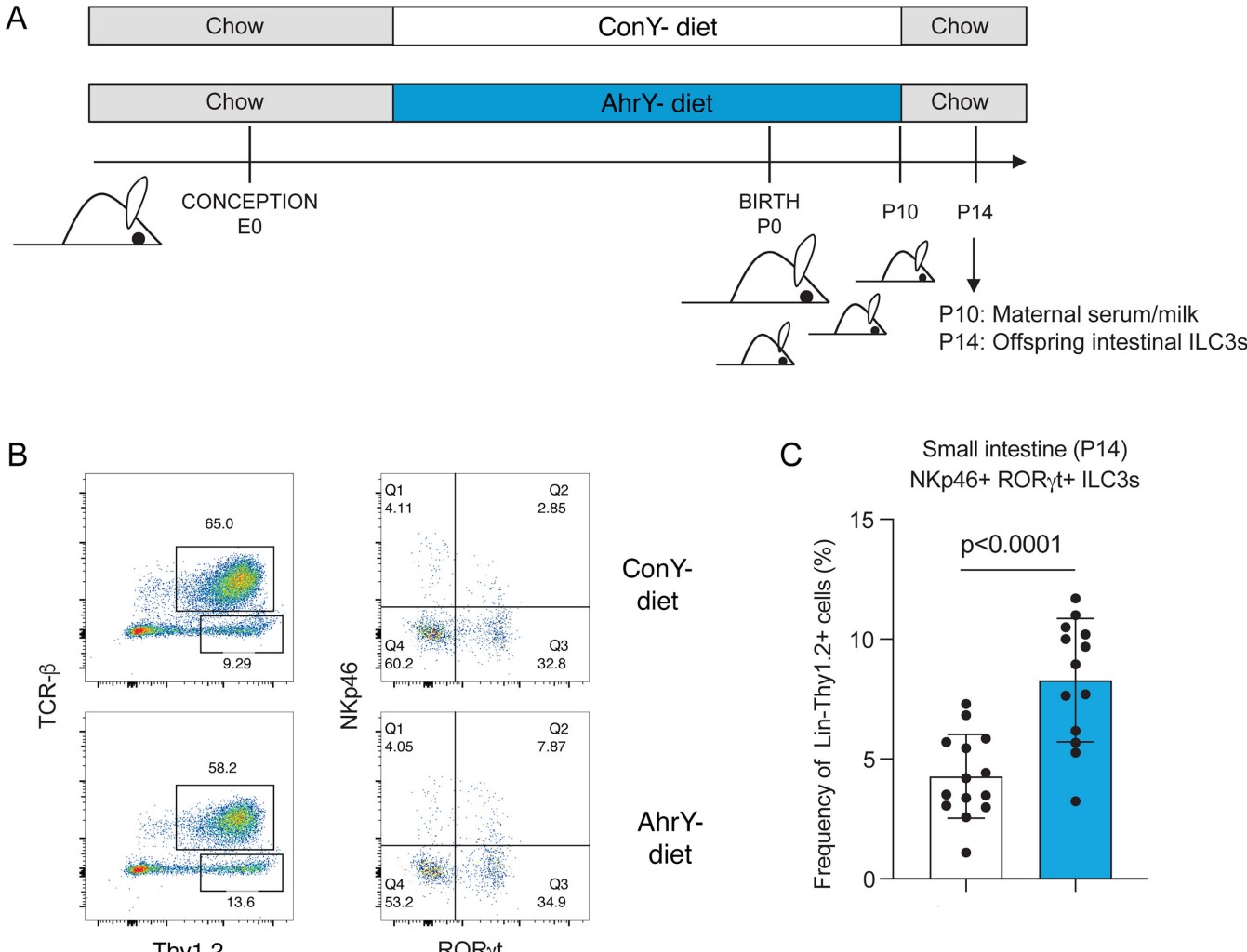

**FIG 3** Feeding germ-free dams with the AhrY-diet increases intestinal group 3 innate lymphoid cells in the offspring. (A) Time pregnant germ-free C57BL/6 dams were fed purified diets containing the AhR yoghurt (AhrY-diet) or the control yoghurt (ConY-diet) starting on day 7 post conception until postnatal day 10. The offspring in each group was analyzed on postnatal day 14 using flow cytometry of the intestinal lamina propria. (B) Representative dot plots pre-gated on CD19$^-$ (left) and CD19$^-$TCRβ$^-$Thy1.2$^+$ (right) small intestinal lamina propria lymphocytes. (C) Relative frequency of small intestinal NKp46$^+$RORgt + ILC3 among CD19$^-$TCRβ$^-$Thy1.2$^+$ at postnatal day 14. Data represent mean ± SD, $n = 14$ pups (ConY-diet), $n = 13$ pups (AhrY-diet) pooled from two independent experiments.

($P$ value = 0.011), glycerophospholipid metabolism ($P$ value = 0.023), and fatty acid metabolism ($P$ value = 0.038) (Fig. S4).

## DISCUSSION

The interplay between diet, the intestinal microbiota, and the mucosal immune system has received increasing attention over the last decades, and it is well appreciated that dietary metabolites can have both direct and indirect, the latter of which are mediated via the commensal microbiota, effects on the host immune system (31). This impact of diet is not only limited to the host itself but seems to even affect the growing child of a pregnant or nursing mammal (32). Breastfeeding versus formula feeding (33, 34), timing of the introduction of solid food (35), and even the maternal diet and microbiota (30, 36) have long-lasting effects on the offspring's microbiota and immune development.

Dietary considerations and the ability to improve the health benefit of foods are, thus, of rising public interest. Nutrition can be altered in several manners. While the simplest, but difficult to achieve, way is to change frequency of consumption of certain foods, the addition of specific metabolites into food products is often necessary. Natural ways of

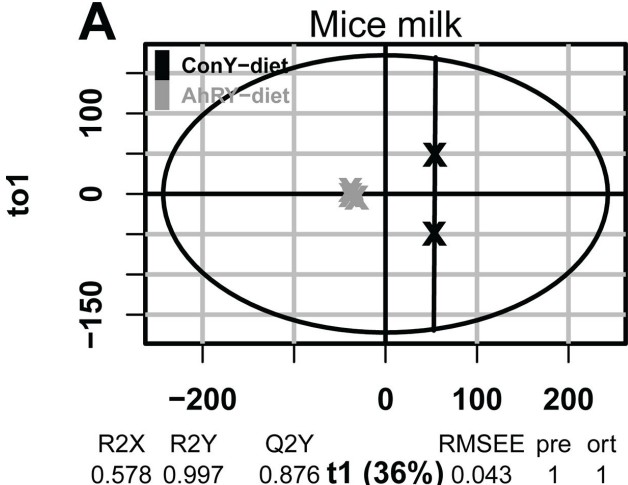

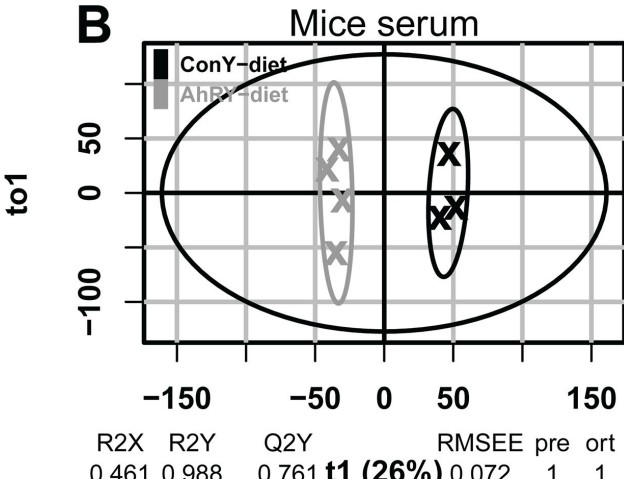

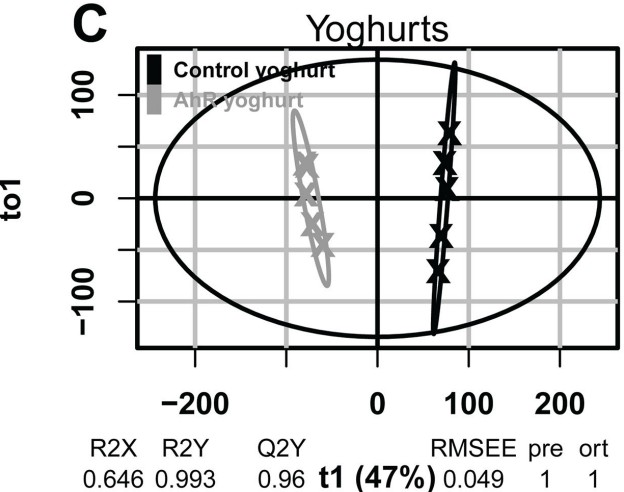

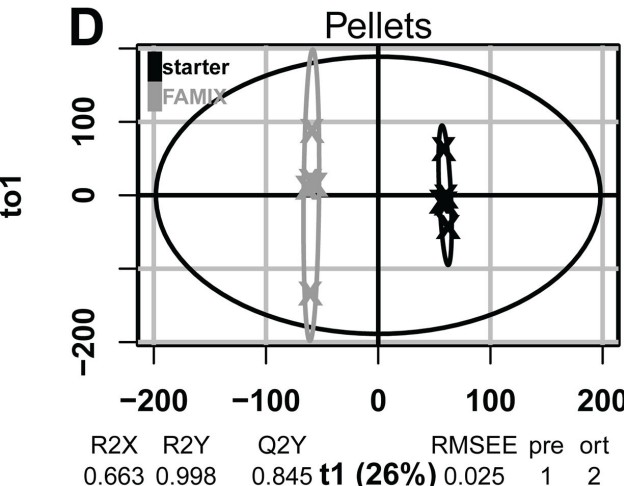

**FIG 4** Untargeted LC-MS metabolomics discriminates the test products from the control products as well as the milk and serum of germ-free dams having ingested them. (A) Orthogonal projections to latent structures discriminant analysis (OPLS-DA) run with 6,662 compounds and discriminating milk from germ-free dams fed the AhrY-diet and ConY-diet. (B) OPLS-DA run with 7,255 compounds and discriminating serum from germ-free dams fed the AhrY-diet and ConY-diet. (C) OPLS-DA run with 12,734 compounds and discriminating the AhR yoghurt from the control yoghurt. (D) OPLS-DA run with 15,271 compounds and discriminating the AhrY-diet from the ConY-diet. Q2Y, predictive ability parameter; R2Y, goodness-of-fit parameter; RMSEE, square root of the mean error between the actual and the predicted responses; pre and ort, number of predictive and orthogonal components respectively; t1, OPLS-DA predictive component; to1, OPLS-DA orthogonal component.

altering nutrient composition, such as by fermentation, has advantages over the addition of artificially produced chemicals. Microbes have an almost endless metabolic capacity and ability to ferment food (18). In addition, fermented foods are usually considered safe, which may especially be an aspect of interest during the sensitive early life period.

Here, we have presented a targeted fermentation approach based on screening genomes of a bacterial collection for genes involved in synthesis of indoles or its derivatives for the fermentation of milk to produce a yoghurt that activates AhR *in vitro*, and with biological effects in the mouse. The pathways leading to the activation of AhR through indoles are complex involving a broad and overlapping array of agonistic and antagonistic metabolites. Our strategy for the genomic selection of bacterial strains therefore remained empirical as primarily relying on the selection of strains with

genomes populated in genes involved in indole metabolism, regardless of metabolic fluxes. Still, we were able to confirm that yoghurt fermented with the routinely used starter culture plus bioinformatically chosen strains was enriched for seven compounds from the tryptophan pathway compared to yoghurt produced with the bacteria present in the starter culture only. Four of these compounds (tryptophol [37], kynurenic acid [38], 5-hydroxy-L-tryptophan [39], and indole-3-pyruvic acid [37]) are known to be AhR agonists and products of microbial metabolism. In particular, the role of tryptophol and indole-3-pyruvic acid in maintaining intestinal epithelial barrier integrity and intestinal immune homeostasis via their interaction with AhR have been described (40).

Among the seven metabolites enriched in the AhR yoghurt, only nicotinic or picolinic acid, and tryptophol were measured as significantly enriched in the AhrY-diet compared to the ConY-diet. This discrepancy could be attributed to several factors. First, the diet used for the incorporation of the yoghurts evidently already contains metabolites of the tryptophan pathway that might influence their relative distribution between the AhrY- and ConY-diets; second, the efficiency of the extraction of metabolites is strongly influenced by the matrix, and this effect might be particularly important when comparing the composition of the dried pellets with the semi-liquid yoghurt; third, as the composition of the yoghurts and murine diets were analyzed on samples stored at −20°C for different time periods, part of the differences in the content of indole derivatives between yoghurt and pellets might be due to different storage times; finally, although irradiation is a well-accepted method to sterilize diets for studies with specific pathogen-free animals, this method also induces changes in the nutritional composition of the diets (41) that may have impacted on the concentration profile of the metabolites of the tryptophan pathway (42). However, taken together, our analytical results are consistent with an increase in metabolites of the tryptophan pathway in the yoghurt and diet prepared with the bacterial strains selected to increase the production of indole derivatives.

The targeted analysis of the panel of 38 indoles was unable to differentiate between the serum and milk samples collected from the dams of the two groups of mice on day 10 post birth, although the yoghurt products could be differentiated. Despite this fact, the concentration of AhR ligands in breast milk may have differed at an earlier time point during lactation when the milk or even the early colostrum had a different composition (43). The experimental set-up presented here was adapted from previous experiments, where germ-free pregnant mice were orally exposed to either a commensal *Escherichia coli* strain producing indoles or to a pure AhR ligand during gestation (30). While we were able to detect AhR ligands in breast milk in the previous experiments, these were measured in maternal milk collected days 1–3 post birth. Collecting milk at this early time point is not possible without contaminating the germ-free animals and was thus not conducted in our experiments where we wanted to analyze intestinal immunity of the pups on postanal day 14 (P14), the earliest day showing a significant difference in NKp46⁺ILC3s in the small intestine of pups born to gestationally colonized versus germ-free dams (30). The impact of AhR ligands in breast milk during the first days post birth may have a long-lasting impact on survival/proliferation of the small intestinal ILC3s as it was also the case in our previous publication (30). The absence of a difference in measurable AhR ligands in the maternal serum may be explained by the high endogenous levels of AhR ligands. An alternative hypothesis includes that certain metabolites specifically accumulate in the mammary glands during pregnancy as lactogenesis starts already before giving birth to the pups (44).

Given the pleotropic activity of downstream signaling of AhR (45), identifying downstream effectors of activation of this receptor is a complex and technically difficult task. To undertake a first step in this direction, we have complemented the analysis of the metabolites of the tryptophan pathway in the milk and serum samples of the dams with an untargeted analysis of their metabolome. A multivariate analysis of the metabolomes revealed a significant difference between the test and control groups in both the milk and serum samples. Interestingly, a pathway analysis with mummichog

identified "prostaglandin formation from arachidonate" as significantly different between the milk metabolome of the mice fed the AhrY-diet compared to the group fed the ConY-diet. This result is interesting in light of the ability of AhR ligands to act as antagonists of the prostaglandin D2 receptor (46). On the other hand, prostaglandins also possess immunomodulatory activities that are independent of AhR ligands (47). In line with these findings, we cannot exclude that the genomic selection of bacteria to produce fermented food matrices with enhanced functionality *in vivo* may extend beyond the targeted endpoint(s) so that fermentation products other than indoles could independently or in synergy contribute to the observed changes in ILC3 numbers. In particular, the potential anti-inflammatory activity of dead bacteria (48), which were evidently not removed from the irradiated pellets, could also be immunologically active. Finally, we note that the mummichog tool used to identify the prostaglandin pathway is based on an enrichment analysis of molecular mases in pathway databases. Although this tool does not directly lead to the identification of metabolites, it nonetheless allows to pinpoint functional properties potentially modulated by the dietary treatment, which could be the target of future research.

As our main scientific question involved feeding pregnant mice with the test yoghurt, it was important for us to circumvent gavaging mice and to incorporate the yoghurt into pellets to minimize the stress the females would be exposed to during gestation and lactation. Importantly, our strategy of freeze drying the yoghurts and incorporating them into chemically defined diets followed by irradiation was successful in producing sterile fermented products that could be ingested by germ-free mice. This approach may be helpful also in future research applications involving similar dietary intervention.

Feeding germ-free dams the AhrY-diet for 24 days during gestation and lactation was sufficient to increase ILC3s in the small intestinal lamina propria of the 14-day-old offspring similar to what we had previously shown when germ-free dams were colonized with a commensal *E. coli strain* or gavaged with high concentrations of a synthetic AhR ligand, indole-3-carbinol (30). In accordance with published data and knowledge in the field of mucosal immunology (24, 30), only offspring intestinal ILC3 expansion and maintenance, but not ILC2 numbers or function, were affected by feeding the AhrY-diet to pregnant and lactating dams. In contrast, both Treg and Th17 cells have been shown to be negatively affected by the absence of AhR (49). However, these T-cell subsets are very scarce in early life (50), and hence, our data are conclusive with existing knowledge on AhR and intestinal immune cell development and homeostasis.

An involvement of the AhR in additional functions of the intestinal immune system was previously shown, such as in the development and maintenance of intraepithelial lymphocytes (25, 51), the release of antimicrobial peptides (24), and the increased differentiation of secretory cells within the intestinal epithelium (52). These data derive from studies in adult mice. Interestingly, maternally orally ingested AhR ligands can reach the breast milk and could activate the AhR in murine small intestinal epithelial cells and protect from necrotizing enterocolitis (NEC), a severe inflammatory condition prevalent in pre-mature babies (53). The same group also showed the presence of AhR ligands in human breast milk samples by testing for AhR-induced gene expression following incubation of WT and *Ahr*−/− intestinal organoid cultures with the human milk samples (53). Although we have not tested if any of these additional effects of AhR, especially within the intestinal epithelium, would also hold true in our experimental setting, additional biological effects of the yoghurt-derived indoles in the breast milk on the maternal and offspring intestinal compartment are likely and would contribute to the health-promoting character of the AhR yoghurt. Being able to boost immune maturation of the newborn child by altering the nutrition of the mother holds promise for preventive or therapeutic measures during the sensitive early life period, also known as the "window of opportunity" during which environmental influences can have long-lasting effects on microbiota and immune maturation (54).

AhR signaling and overexpression has originally been associated with several tumor types. Considering the ubiquitous expression of the AhR and its role in proliferation and

reaeration of cells, this is not surprising (55). It is, however, nowadays accepted that AhR has important endogenous functions, which are especially important for metabolization of xenobiotics in the liver and also in the intestinal immune system as mentioned earlier. AhR signaling may be dependent on several factors, such as the affinity of the ligand to the receptor, the expression of the AhR repressor, and additional intracellular co-factors that can all lead to a cell-type and situation-specific expression profile of the AhR-target genes. Important differences also exist between the AhR ligand signaling induction in mice and humans (56), and this may be considered when designing follow-up human studies. The fact that we have used a human hepatocyte cell line to screen the original set of yoghurts for their ability to activate the AhR *in vitro* holds promise for cross-species applicability of our *in vivo* result. Our approach of targeted fermentation of milk to produce a yoghurt rich in indole derivatives was successful and seems a promising tool that may be applied to introduce naturally produced metabolites into fermented foods and is in line with the global debate on the reintroduction of safe microbes into industrialized food to improve overall nutrition (57).

In conclusion, this report shows that the selection of LAB enriched in genes populating pathways involved in indole metabolism can lead to the production of a yoghurt product that, when fed to pregnant mice, activates the intestinal immunity of their offspring. Although the composition of the yoghurt products and a functional cellular gene expression assay supported the role of indole derivatives and AhR activation in mediating these effects, an analysis of the metabolome of the products and murine samples indicated broader signaling mechanisms. Our findings thus raise interesting questions that should motivate future studies in animal models as well as in humans.

## MATERIALS AND METHODS

### Genomic selection of bacterial strains

To select lactic acid bacteria with the potential to synthesize AhR-activating ligands, we annotated the genomes of 631 strains from the Agroscope strain collection using the KEGG Automatic Annotation Server (KAAS) (58). This enabled the identification of key genes within the phenylalanine, tyrosine, and tryptophan biosynthesis pathways. The data were integrated into a web-app, the precursor of OpenGenomeBrowser (59). It displayed the key genes present in each strain and their respective copy number, and grouped strains with an identical coverage pattern together. For each group, the web-app visualized the pathway coverage using the KEGG website, generated a phylogenetic tree of the genomes based on whole-genome OrthoANI-similarity (v. 1.40) (60), and calculated a protein sequence alignment for each of the key genes using Clustal Omega (v. 1.2.4) (61). The resulting sequence alignment was then visualized using MView (v. 1.64) (62). Subsequently, a phylogenetic tree based on this alignment was calculated using IQ-TREE (v. 1.6.10) (63).

The selection was based on (i) genetic coverage of the pathways of interest, including reactions that branch off; (ii) phylogenetic relatedness; (iii) gene variants; (iv) gene copy numbers; and (v) genetic coverage of other annotations, such as antibiotic resistance or gut adherence genes. Thus, we selected 125 strains, spanning seven different genera of LAB, for subsequent yogurt production and phenotypic characterization.

### Yoghurt production

All 125 test yoghurts were produced at Agroscope (Bern) to industrial standards in accordance with Swiss food legislation. Lactose-free, full-fat (3.5%), homogenized, pasteurized cow milk (Aha! IP Suisse, Migros, Switzerland) was used. The 125 tested strains were precultured 16 to 24 h at 30°C or 37°C in their respective growth medium, before being added to milk along with a classical yoghurt starter culture consisting of a mix of *Lactobacillus delbrueckii* subsp. *bulgaricus* and *Streptococcus salivarius* subsp. *thermophilus* (Yoflex YC-381, Chr. Hansen A/S, Denmark). One yoghurt, the control

yoghurt, contained only the starter culture without any additional strain. Milk was fermented during 16 h at 37°C, cooled down to 4°C and stored at – 20°C prior to analysis.

### In vitro AhR reporter system

The ability of the test yoghurts to activate the AhR promoter was evaluated *in vitro* using the HepG2–AhR–Luc cell line. Human HepG2 liver carcinoma AhR–Lucia reporter (HepG2–Lucia AhR) cells were purchased from InvivoGen and grown in Eagle's minimal essential medium (EMEM, ThermoFisher), supplemented with 10% (vol/vol) heat-inactivated (30 min at 56°C) fetal bovine serum (ThermoFisher), 1× non-essential amino acids (ThermoFisher), 100 U/mL of Pen-Strep (ThermoFisher), and 100 µg/mL of Normocin (InvivoGen).

At the start of the assay, cells were rinsed with phosphate-buffered saline (PBS), detached with trypsin, centrifuged (300 $g$, 5 min, 20°C), counted, and resuspended 1.1 × $10^5$ cells/mL in test medium.

Of the yoghurt sample, 20 µL was added to a well of a flat-bottom 96-well plate. An AhR agonist (e.g., 6-formylindolo(3,2-b)carbazole in EMEM at 0.05, 0.5, and 5 µg/mL of final concentrations) was used as the positive control and endotoxin-free water as the negative control. Of the cell suspension (~20,000 cells) per well, 180 µL was added to each well and incubated at 37°C in a $CO_2$ incubator for 72 h. Of the HepG2-Lucia AhR stimulated cell supernatant, 40 µL was transferred into a 96-well white (opaque) or black plate. Of the QUANTI-Luc solution, 100 µL was added quickly to all wells before immediate measurement with the TECAN Reader Infinite 200 [reading time (integration time) was 500 ms, and the end point measurement (settle time) was set at 1,000].

### Production of the yogurt-based pellets for the mice studies

Two yoghurts were prepared at Agroscope for the *in vivo* studies. The control yoghurt, containing only the starter strains, and the AhR yoghurt containing the starter and two additional strains selected from the 125 previously tested *in vitro*. The yoghurts were lyophilized and shipped to Research Diets, Inc. (New Brunswick, NJ, USA) where they were incorporated at 40% (wt/wt) into an open standard diet (diet D11112201 for mice). The pellets were sterilized by two rounds of gamma-ray irradiation (10–20 kGy) and one round of X-ray irradiation (30–60 kGy). They were imported into a surgical isolator after disinfection of the bag with 2% peracetic acid. The pellets containing the AhR yoghurt are referred to as the AhrY-diet, whereas the pellets containing the control yoghurt are referred to as the ConY-diet. The sterility of the diets was assessed prior to each experiment by aerobic and anaerobic cultures and Sytox and Gram stainings of the cecal contents of germ-free mice that were fed the purified diets for 7 days and switched back to autoclaved chow for another 14 days.

### In vivo studies

Germ-free (GF) C57BL/6 were bred and maintained in flexible-film isolators at the Clean Mouse Facility, University of Bern, Switzerland, as previously described (64). Germ-free status was routinely monitored by culture-dependent and -independent methods. All mouse experiments were performed in accordance with Swiss Federal and Cantonal regulations under the cantonal license BE104/20. Mice were born and raised while fed conventional chow (3307, Kliba Nafag).

GF females at the age of 8–10 weeks were time-mated (vaginal plug check) in experimental sterile isolators and switched to either the AhrY-diet or ConY-diet 7 days post mating and kept on the experimental diet until 10 days post giving birth. Mice were then switched back to chow to prevent the growing pups from being exposed to yoghurt-containing diets that may have fallen into the cage. Dams and offspring were exported from the experimental isolator on postnatal day 14 and analyzed as described. As the earliest day showing a significant difference in NKp46[+]ILC3 numbers in the small intestine of pups born to gestationally colonized versus germ-free dams was day 14 in

the study of Gomez de Agüero and colleagues (30), we also performed the tissue analysis on postnatal day 14. Sterility of the mice was confirmed continuously throughout the experiment by culture-dependent and -independent methods.

## Isolation of small intestinal lamina propria lymphocytes

The intestines were removed from the mouse and placed in ice-cold Dulbecco's phosphate-buffered saline (DPBS; Gibco). Residual fat and Peyer's patches were removed. The intestine was opened longitudinally before cutting into 2-cm segments. The tissue was washed once in ice-cold DPBS followed by four washes of 8 min in 15 mL of DPBS with 5 mM EDTA and 10 mM HEPES with shaking at 37°C to remove epithelial cells. Residual tissue was then washed in 15 mL of Iscove's modified Dulbecco's medium (IMDM) containing 10% fetal calf serum (FCS) (IMDM/FCS) at 37°C for 8 min before being minced and digested in 15 mL of IMDM containing 0.5 mg/mL of collagenase type VIII (Sigma) and 10 U/mL of DNase I (Roche) with shaking at 37°C for 20–30 min (small intestine) or 30–40 min (colon). The obtained cell suspension was passed through a cell strainer (100 µm) and washed with IMDM/FCS. Cells were centrifuged (600 $g$, 7 min, 4°C) and resuspended in FACS buffer (PBS, 2% FCS, 2 mM EDTA, 0.01% NaN$_3$) for staining for flow cytometry analysis.

## Flow cytometry

Cells were washed once with DPBS before being stained with fixable viability dye (eBioscience) and anti-mouse-CD16/CD32 Fc-receptor block (93, Biolegend) diluted in DPBS for 30 min on ice. Single-cell suspensions were sequentially incubated with primary/biotin- and fluorescence-coupled antibodies diluted in FACS buffer for 15 min on ice. For intracellular stainings, cells were fixed and permeabilized using the Transcription Factor Staining Buffer Set (eBioscience). Antibodies for intracellular staining were diluted in the permeabilization buffer from the Transcription Factor Staining Buffer Set and incubated at 4°C overnight. The following mouse-specific conjugated antibodies were used: CD19 (6D5, Biolegend), CD4 (RM4-5, Biolegend), CD44 (IM7, Biolegend), CD62L (MEL-14), CD8a (53–6.7, BD Bioscience), Foxp3 (FJK-16s, ThermoFisher), Gata-3 (TWAJ, Biolegend), Helios (22F6, Biolegend), NKp46 (19A1.4, Biolegend), RORγt (B2D, ThermoFisher), T-bet (4B10, ThermoFisher), TCR-β (H57-597, Biolegend), TCR-gd (GL3, BD Bioscience), and Thy1.2 (53–2.1, Biolegend). Data were acquired on an LSRFortessa (BD Biosciences) and analyzed using FlowJo software version 10.6.2 (Tree Star Inc.). In all experiments, FSC-H versus FSC-A was used to gate on singlets using dead cells exclusion. Where lineage exclusion was performed, TCR- and CD19-expressing cells were removed from further analysis.

## Collection of breast milk

Ten days after delivery, the nursing dams were separated from their pups for 4 h before milking. Dams were anesthetized with isoflurane according to standard operating procedures, and 1 U of oxytocin (Syntocinon) was injected intraperitoneally. The collection of breast milk was started within 5 min using a custom-made vacuum pump-based collection device. Aliquots of milk were frozen in liquid nitrogen and stored at −80°C until further analysis.

## UHPLC-MS metabolomics analysis

Protein precipitation of murine serum (postnatal day 10), murine milk (postnatal day 10), and food products (milk, control yoghurt, AhR yoghurt and pellets) was obtained with the addition 1:4 (vol/vol) of acetonitrile containing 1% (vol/vol) formic acid. Samples were homogenized by vortexing (murine serum, murine breast milk, milk, and yoghurt samples) or using the Retsch MM300 Mixer at 30 Hz for 2 min (pellets). Centrifugation (12,000 rpm for 15 min at 4°C) and filtration of the supernatant through a 0.22-µm regenerated cellulose filter (WhatmanTM UnifloTM 13/0.2 RC) was performed for the

food product samples. Protein-free extracts were then filtered through a phospholipid filter membrane to limit ion suppression (Phree, Phenomenex Inc., Torrance, California, USA). The filtrate was then injected into the UHPLC/MS system consisting of an UltiMate 3000 HPLC (Thermo Fisher Scientific) coupled to a maXis 4G + quadrupole time-of-flight mass spectrometer (MS) with electrospray interface (Bruker Daltonik GmbH, Bremen, Germany). Chromatographic separation was performed on a C18 hybrid silica column (Acquity UPLC HSS T3 1.8 µm 2.1 × 150 mm, Waters, UK), reversed phase at a flow rate of 0.4 mL/min. The mobile phase consisted of ultrafiltered water (Milli-Q IQ 7000, Merck, Germany) containing 0.1% formic acid (FlukaTM, Honeywell, USA) (A), and acetonitrile (Supelco, Merck, Germany) with 0.1% formic acid (B), with the following elution gradient (A:B): 95:5 at 0 min to 5:95 at 10 min; 5:95 from 10 to 20 min; 95:5 from 20 to 30 min for untargeted metabolomics. The spectra were recorded from m/z 75 to m/z 1,500 in positive ion mode. Detailed MS settings were as follows: collision-induced dissociation: 20 to 70 eV, electrospray voltage: 4.5 kV, endplate offset: 500 V, capillary voltage: 3,400 V, nitrogen flow: 4 mL/min at 200°C, spectra acquisition rate: 1 Hz in profile mode, resolution: 80,000 FWHM.

The extracts from pellets were analyzed using another UHPLC/MS system consisting of an Acquity UHPLC system equipped with a C18 hybrid silica column (Acquity UPLC HSS T3 1.8 µm 2.1 × 100 mm, Waters, UK), using the same gradient and mobile phases as described above and coupled to a G2-XS QTOF mass spectrometer equipped with an electrospray source (Waters AG, Switzerland). The spectra were recorded from m/z 75 to m/z 1,500 in positive ion mode. Detailed MS settings were as follows: the data were acquired with scans of 0.1 s with a collision energy ramp from 20 to 70 V. The capillary and cone voltages were set to 2.0 kV and 20 V, respectively. The source temperature was maintained at 140°C, the desolvation was 400°C at 1,000 L/h, and cone gas flow was 100 L/h. Accurate mass measurements (<2 ppm) were obtained by infusing a solution of leucin encephalin at 200 ng/mL and a flow rate of 10 mL/min through the Lock Spray probe. The identification of the metabolites was obtained by comparing mass spectra and retention times with those of pure standards.

Untargeted mass spectrometry data were processed using Progenesis QI (v.2.3.6198.24128; NonLinear Dynamics Ltd.) for retention time correction, peak-picking, deconvolution, adduct annotation, and normalization (default automatic sensitivity and without minimum peak width).

Orthogonal partial least-square discriminant analysis (OPLS-DA) of the product metabolomes was performed using R (v.4.2.3; R Foundation for Statistical Computing, Vienna, Austria) and ROPLS package (v1.34.0). Validity of the models was evaluated by the goodness-of-fit parameter (R2Y), the predictive ability parameter (Q2; calculated by sixfold cross-validation, Q2 >0.50 as a cutoff value), and permutation tests with 500 random permutations to exclude any random separation of the sample groups. Analysis of the pathways associated with the differentially expressed metabolic features was conducted with mummichog (tool available in MetaboAnalyst V6.0), a computational algorithm, which predicts functional activity directly from spectral features without *a priori* identification of metabolites (65).

The targeted analysis of the yoghurts and pellets focused on 38 metabolites of the tryptophan pathway and, in particular, indole derivatives (Table S1). The analysis was performed as described above for untargeted metabolomics except that the liquid chromatography gradient was (A:B): 95:5 at 0 min to 30:70 at 15 min; 5:95 at 16 min until 20 min; and 95:5 from 20 to 30 min. The presence of the metabolites in the samples was investigated using pure standards (Table S1). The relative amount of each compound was assessed using the signal intensity given by Bruker Compass Data Analysis (Bruker Daltonik, GmbH). Intensities between control yoghurt and AhR yoghurt were compared by a Wilcoxon test ($P < 0.05$ as significance threshold) using R.

## ACKNOWLEDGMENTS

We are grateful for the support by the Clean Mouse Facility, which is supported by the Genaxen Foundation, Inselspital, and the University of Bern. We would like to thank Nerea Fernadez Trigo (University of Bern) for technical help and Andrew J. Macpherson (University of Bern) for critically reviewing the manuscript. This work was supported by a grant from the Gebert Rüf foundation (Polyfermenthealth—Microbials 2017) to G.V. and A.J.M. S.C.G.V. was supported through a Peter Hans Hofschneider Professorship provided by the Stiftung Molekulare Biomedizin and an SNSF project grant (310030_212511).

## AUTHOR AFFILIATIONS

[1]Agroscope, Schwarzenburgstrasse, Bern, Switzerland

[2]Interfaculty Bioinformatics Unit, University of Bern, Bern, Switzerland

[3]Department of Visceral Surgery and Medicine, Inselspital, Bern University Hospital, Bern, Switzerland

[4]Department for BioMedical Research (DBMR), University of Bern, Bern, Switzerland

[5]Institute of Plant Sciences, University of Bern, Switzerland, Bern

## AUTHOR ORCIDs

Zahra Sattari http://orcid.org/0000-0002-2102-4031

Christelle A. M. Robert http://orcid.org/0000-0003-3415-2371

Rémy Bruggmann http://orcid.org/0000-0001-5629-6363

Stephanie C. Ganal-Vonarburg http://orcid.org/0000-0002-2548-7754

Guy Vergères http://orcid.org/0000-0003-4574-0590

## FUNDING

| Funder | Grant(s) | Author(s) |
| --- | --- | --- |
| Gebert Rüf Stiftung (Gebert Rüf Foundation) | GRS-070/17 | Thomas Roder |
| Gebert Rüf Stiftung (Gebert Rüf Foundation) | GRS-070/17 | Zahra Sattari |
| Swiss National Science Foundation | 310030_212511 | Stephanie C. Ganal-Vonarburg |
| MD-PhD scholarship of the Swiss National Science Foundation | 323530_199385 | Stephanie C. Ganal-Vonarburg |

## AUTHOR CONTRIBUTIONS

Grégory Pimentel, Conceptualization, Investigation, Writing – original draft, Writing – review and editing | Thomas Roder, Conceptualization, Formal analysis, Investigation, Methodology, Software, Writing – original draft, Writing – review and editing | Cornelia Bär, Conceptualization, Formal analysis, Investigation, Supervision, Writing – original draft, Writing – review and editing | Sandro Christensen, Formal analysis, Investigation, Writing – review and editing | Zahra Sattari, Formal analysis, Investigation, Writing – review and editing | Cristina Kalbermatter, Investigation, Writing – review and editing | Ueli von Ah, Conceptualization, Formal analysis, Investigation, Writing – review and editing | Christelle A. M. Robert, Investigation, Writing – review and editing | Pierre Mateo, Investigation, Writing – review and editing | Rémy Bruggmann, Conceptualization, Investigation, Supervision, Writing – review and editing | Stephanie C. Ganal-Vonarburg, Conceptualization, Investigation, Supervision, Writing – original draft, Writing – review and editing | Guy Vergères, Conceptualization, Funding acquisition, Investigation, Project administration, Supervision, Writing – original draft, Writing – review and editing

## ADDITIONAL FILES

The following material is available online.

### Supplemental Material

**Figure S1 (Spectrum00393-24-S0001.pdf).** In vitro AhR activation assay of 125 test yoghurts as compared to the conventional control yoghurt.
**Figure S2 (Spectrum00393-24-S0002.pdf).** Feeding germ-free dams with the AhrY-diet.
**Figure S3 (Spectrum00393-24-S0003.pdf).** OPLS-DA permutation tests plots.
**Figure S4 (Spectrum00393-24-S0004.pdf).** Functional analysis of compounds masses detected in murine milk.
**Supplemental material (Spectrum00393-24-S0005.pdf).** Legends for supplemental figures and tables.
**Table S1 (Spectrum00393-24-S0006.docx).** List of tryptophan derivatives targeted by UHPLC-MS in the dairy products as well as mice samples.
**Table S2 (Spectrum00393-24-S0007.xlsx).** Macro- and micro-nutrient composition of the yoghurt-containing diets.

### Open Peer Review

**PEER REVIEW HISTORY (review-history.pdf).** An accounting of the reviewer comments and feedback.

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
