## [Reviewer comments · Microbiology Spectrum]

Microbiology Spectrum

Maternal consumption of yoghurt activating the aryl hydrocarbon receptor increases group 3 innate lymphoid cells in murine offspring.

Grégory Pimentel, Thomas Roder, Cornelia Bär, Sandro Christensen, Zahra Sattari, Cristina Kalbermatter, Ueli von Ah, Christelle Robert, Pierre Mateo, Rémy Brugmann, Stephanie Ganal-Vonarburg, and Guy Vergeres

Corresponding Author(s): Guy Vergeres, Agroscope

Review Timeline:

Submission Date:	February 12, 2024
Editorial Decision:	March 21, 2024
Revision Received:	August 23, 2024
Editorial Decision:	September 9, 2024
Revision Received:	September 25, 2024
Accepted:	September 25, 2024

Editor: Laxmi Yeruva

Reviewer(s): Disclosure of reviewer identity is with reference to reviewer comments included in decision letter(s). The following individuals involved in review of your submission have agreed to reveal their identity: Manoj Gurung (Reviewer #1)

Transaction Report:

DOI: <https://doi.org/10.1128/spectrum.00393-24>

Re: Spectrum00393-24 (Maternal consumption of yoghurt activating the aryl hydrocarbon receptor increases group 3 innate lymphoid cells in murine offspring.)

Dear Dr. Guy Vergeres:

Thank you for the privilege of reviewing your work. Below you will find my comments, instructions from the Spectrum editorial office, and the reviewer comments.

Revision Guidelines

Sincerely,
Laxmi Yeruva
Editor
Microbiology Spectrum

Reviewer #2 (Comments for the Author):

In this paper, the LAB that can activate AhR were screened to ferment yogurt. The fermented product was added to the diet of pregnant dams to analyze its effect on the innate immune cells of offspring intestine. The article is well written. There are some minor questions.

1 The study lacks the ingredients of the animal diets. What is the amount of indole metabolites in the AhrY-diet?

2 Dams are dietary intake of AhrY-diet containing indole metabolites, what are the concentrations of these metabolites in Dam's serum and breast milk? So the pathways that these metabolites affect the intestine innate immune could be analyzed.

3 The biological significance of the AhrY-diet having an effect only on specific to ILC3s Instead of ILC2 and T cell needs to be discussed in the discussion.

Thank you for the opportunity to review the manuscript “Maternal consumption of yogurt activating the aryl hydrocarbon receptor increases group 3 innate lymphoid cells in murine offspring. In this manuscript, the investigators screened the lactic acid bacteria for their potential to produce aryl hydrocarbon receptor ligands, especially indole compounds, in vitro. They used the yogurt fermented by selected LAB in pregnant germ-free mice to study the intestinal immune innate lymphoid cells. The investigators reported that the chosen yogurt, when incorporated into mice diets in lyophilized form and fed to germ-free pregnant mice, increased the ILC3 in the small intestinal lamina propria of the offspring. The investigators’ approach to incorporating freeze-dried fermented products into a chow diet and successful feeding in germ-free pregnant mice and studying the impact on offspring potentially promotes future studies involving maternal dietary intervention. I find the manuscript well-written. However, I have some major comments on the manuscript.

Major Comments:

- 1) Though the investigators showed a clear difference in ILC3 in small intestine lamina propria lymphocytes between the two diet groups, the mechanism behind this effect in offspring is unclear. No difference in AhR-activating metabolites was found in milk or maternal serum. Hence, I wonder if the effect is due to the AhR ligands in the AhrY diet. Did the investigators evaluate the AhR-associated ligand in these diets? What is the impact of lyophilization and irradiation on the AhR ligands? If there are no differences in LAB-associated AhR-ligands between the two diets, the overall conclusion of the manuscript will be different. The authors did agree that besides indole and its derivatives, other potential factors (like anti-inflammatory factors) could contribute to such an effect in offspring. Since indoles were not detected or were not different between the two groups in serum and milk, the role of other factors is more plausible.
- 2) Untargeted metabolomics of milk and maternal serum showed a separation between the AhRY and ConY diets. However, which metabolomes induce this difference between the two groups has not been reported. The major metabolome that discriminates these groups may be impacting the offspring ILC3 cell population. I would suggest the investigators explore the metabolome data and report which metabolomes are contributing to the difference.
- 3) Did the investigators measure any ILC3-related cytokines, like IL-22? It will be interesting to know whether the innate lymphoid cell population increase reflects the related cytokine production.
- 4) What was the rationale of tissue harvesting on day 14 when mice were changed to a regular chow diet? It is unclear why the investigators did not harvest tissues/serum/milk on day 10 when mice were still on the AhrY diet.

Minor comments: none

Point-by-Point Response to the reviewer comments on:

"Maternal consumption of yoghurt activating the aryl hydrocarbon receptor increases group 3 innate lymphoid cells in murine offspring." by Grégory Pimentel, Thomas Roder, Cornelia Bär, Sandro Christensen, Zahra Sattari, Ueli von Ah, Rémy Brugmann, Stephanie Ganal-Vonarburg, and Guy Vergeres [Paper #Spectrum00393-24]

Reviewer #1 (Comments for the Author)

Thank you for the opportunity to review the manuscript "Maternal consumption of yogurt activating the aryl hydrocarbon receptor increases group 3 innate lymphoid cells in murine offspring. In this manuscript, the investigators screened the lactic acid bacteria for their potential to produce aryl hydrocarbon receptor ligands, especially indole compounds, in vitro. They used the yogurt fermented by selected LAB in pregnant germ-free mice to study the intestinal immune innate lymphoid cells. The investigators reported that the chosen yogurt, when incorporated into mice diets in lyophilized form and fed to germ-free pregnant mice, increased the ILC3 in the small intestinal lamina propria of the offspring. The investigators' approach to incorporating freeze-dried fermented products into a chow diet and successful feeding in germ-free pregnant mice and studying the impact on offspring potentially promotes future studies involving maternal dietary intervention. I find the manuscript well-written. However, I have some major comments on the manuscript.

Comment 1a

Though the investigators showed a clear difference in ILC3 in small intestine lamina propria lymphocytes between the two diet groups, the mechanism behind this effect in offspring is unclear. No difference in AhR-activating metabolites was found in milk or maternal serum. Hence, I wonder if the effect is due to the AhR ligands in the AhrY diet.

Response

The reviewer's comment is well justified and we have added an additional paragraph in the Discussion on pages 14-15 to present and possibly explain this misalignment in the results:

“The targeted analysis of the panel of 38 indoles was unable to differentiate between the serum and milk samples collected from the dams of the two groups of mice on day 10 post birth although the yoghurt products could be differentiated. Despite this fact, the concentration of AhR ligands in breast milk may have differed at an earlier time point during lactation when the milk or even the early colostrum had a different composition⁽⁴³⁾. The here presented experimental set-up was aligned with experiments performed previously, where germ-free pregnant mice were orally exposed to either a commensal *E. coli* strain producing indoles or to a pure AhR ligand during gestation⁽³⁰⁾. While we were able to detect AhR ligands in breast milk in the previous experiments, these were measured in maternal milk collected day 1-3 post birth. Collecting milk at this early time point is not possible without contaminating the germ-free animals and was thus not conducted in our experiments where we wanted to analyze intestinal immunity of the pups on postnatal day 14 (P14), the earliest day showing a significance difference in NKp46+ ILC3s in the small intestine of pups born to gestationally colonized versus germ-free dams⁽³⁰⁾. The impact of AhR ligands in breast milk during the first days post birth may have a long-lasting impact on survival/proliferation of the small intestinal ILC3s as it was also the case in our previous publication⁽³⁰⁾. The absence of a difference in measurable AhR ligands in the maternal serum may be explained by the high endogenous levels of AhR ligands. An alternative hypothesis includes that certain metabolites specifically accumulate in the mammary glands during pregnancy as lactogenesis starts already before giving birth to the pups⁽⁴⁴⁾.”

References

30. Gomez de Agüero M, Ganal-Vonarburg SC, Fuhrer T *et al.* (2016) The maternal microbiota drives early postnatal innate immune development. *Science* **351**, 1296-1302.
43. Ballard O, Morrow AL (2013) Human milk composition: nutrients and bioactive factors. *Pediatr Clin North Am* **60**, 49-74.
44. Pillay J, J. DT (2024) *Physiology, Lactation*: StatPearls Publishing.

Comment 1b

Did the investigators evaluate the AhR-associated ligand in these diets? What is the impact of lyophilization and irradiation on the AhR ligands?

Response

We have now addressed the first question in the results section “Production and composition of the purified diets containing yoghurt for mice” on pages 9-10 as follows:

“Among the eight indole derivatives differing in the yoghurts, tryptophol and nicotinic or picolinic acid were significantly higher in the AhrY-diet compared to the ConY-diet (Fig. 2J-2K). Significant differences in tryptophan were not measured (Fig. 2I) and the remaining five metabolites could not be detected in the murine diets. These findings confirmed the selection of the AhR- and control yoghurts for formulation of the murine diets for functional tests.”

Furthermore, we have now addressed the first and second questions in the Discussion on pages 13-14 as follows:

“Among the seven metabolites enriched in the AhR yoghurt only nicotinic or picolinic acid, and tryptophol were measured as significantly enriched in the AhrY-diet compared to the ConY-diet. This discrepancy could be attributed to several factors. Firstly, the diet used for the incorporation of the yoghurts evidently already contains metabolites of the tryptophan pathway what might influence their relative distribution between the AhrY- and ConY-diets; secondly, the efficiency of the extraction of metabolites is strongly influenced by the matrix and this effect might be particularly important when comparing the composition of the dried pellets with the semi-liquid yoghurt; thirdly, as the composition of the yoghurts and murine diets were analyzed on samples stored at -20°C for different time periods, part of the differences in the content of indole derivatives between yoghurt and pellets might be due to different storage times; finally, although irradiation is a well-accepted method to sterilize diets for studies with specific pathogen-free animals, this method also induces changes in the nutritional composition of the diets⁽⁴¹⁾ what may have impacted on the concentration profile of the metabolites of the tryptophan pathway⁽⁴²⁾. However, taken together, our analytical results are consistent with an increase in metabolites of the tryptophan pathway in the yoghurt and diet prepared with the bacterial strains selected to increase the production of indole derivatives.”

References

41. Caulfield CD, Cassidy JP, Kelly JP (2008) Effects of gamma irradiation and pasteurization on the nutritive composition of commercially available animal diets. *J Am Assoc Lab Anim Sci* 47, 61-66.
42. Takano Y, Kaneko T, Kobayashi K et al. (2004) Experimental verification of photostability for free- and bound-amino acids exposed to γ -rays and UV irradiation. *Earth, Planets and Space* 56, 669-674.

Note also that the details of the methods used to extract (pages 24-25) and analyze (pages 25-26) the pellets have been added in different parts of the section "UHPLC-MS metabolomics analysis" of the Material and Method.

Comment 1c

If there are no differences in LAB-associated AhR-ligands between the two diets, the overall conclusion of the manuscript will be different. The authors did agree that besides indole and its derivatives, other potential factors (like anti-inflammatory factors) could contribute to such an effect in offspring. Since indoles were not detected or were not different between the two groups in serum and milk, the role of other factors is more plausible.

Response

The differences observed for tryptophol and nicotinic or picolinic acid in the murine diet mitigates the reviewer's comment (see new panels J/K in Figure 2 and response to comment 1b).

In addition, as now explained on page 14-15 of the Discussion, in response to the reviewer's comment 1a, differences in AhR ligands in the mice milk may have been missed due to the timing of the milk collection. Also, a long-term impact on the survival/proliferative capacity of the ILC3s in the small intestine may still be present as observed in our original report (Gomez de Agüero et al., *Science*, 2016).

Finally, we note to the intention of Reviewer, that we did not observe a difference in total AhR ligands in serum in our original report as well as in other subsequent studies. This point has also been addressed in the Discussion in response to comment 1a of Reviewer.

Comment 2

Untargeted metabolomics of milk and maternal serum showed a separation between the AhrY and ConY diets. However, which metabolomes induce this difference between the two groups has not been reported. The major metabolome that discriminates these groups may be impacting the offspring ILC3 cell population. I would suggest the investigators explore the metabolome data and report which metabolomes are contributing to the difference.

Response

We have now addressed the data derived from the analysis of the metabolomes of the murine milk and serum samples in more details as shown below.

Revision of paragraph "The composition of milk and serum of germ-free dams fed the AhrY-diet differs from the one of dams fed the ConY-diet" on page 11 of Results section:

"On postnatal day 10, serum and breast milk were collected and the samples were analyzed by mass spectrometry for indole derivatives and their untargeted metabolome. No significant differences in the relative concentration of the indole derivatives were observed between the experimental groups. However, the untargeted multivariate analysis of the metabolomes (Fig. 4 and Fig. S3) could clearly differentiate serum (A) and breast milk (B) of dams fed the AhrY-diet from dams fed the ConY-diet. The untargeted metabolome analysis could also differentiate the tested foods, i.e., the AhR yoghurt from the control yoghurt (C) as well as the AhrY-diet from the ConY-diet (D).

"Pathway analysis of mice milk and serum metabolomes using the mummichog tool revealed four pathways differentiating murine milk from AhrY-diet fed dams from control dams: linoleate metabolism

(p value = 8.71×10^{-5}), prostaglandin formation from arachidonate (p value = 0.011), glycerophospholipid metabolism (p value = 0.023) and, fatty acid metabolism (p value = 0.038).”

In the Discussion section, we have added the following paragraph on pages 15-16:

“Given the pleiotropic activity of downstream signaling of AhR⁽⁴⁵⁾, identifying downstream effectors of activation of this receptor is a complex and technically difficult task. To undertake a first step in this direction we have complemented the analysis of the metabolites of the tryptophan pathway in the milk and serum samples of the dams with an untargeted analysis of their metabolome. A multivariate analysis of the metabolomes revealed a significant difference between the test and control groups in both the milk and serum samples. Interestingly, a pathway analysis with mummichog identified ‘prostaglandin formation from arachidonate’ as significantly different between the milk metabolome of the mice fed the AhrY-diet compared to the group fed the ConY-diet. This result is interesting in light of the ability of AhR ligands to act as antagonists of the prostaglandin D2 receptor⁽⁴⁶⁾. On the other hand, prostaglandins also possess immunomodulatory activities that are independent of AhR ligands⁽⁴⁷⁾. In line with these findings, we cannot exclude that the genomic selection of bacteria to produce fermented food matrices with enhanced functionality in vivo may extend beyond the targeted endpoint(s) so that fermentation products others than indoles could independently or in synergy contribute to the observed changes in ILC3 numbers. In particular, the potential anti-inflammatory activity of dead bacteria⁽⁴⁸⁾, which were evidently not removed from the irradiated pellets, could also be immunologically active. Finally, we note that the mummichog tool used to identify the prostaglandin pathway is based on an enrichment analysis of molecular masses in pathway databases. Although this tool does not directly lead to the identification of metabolites, it nonetheless allows to pinpoint functional properties potentially modulated by the dietary treatment, which could be the target of future research.”

As well as the following conclusion on page 18-19 of the Discussion:

“In conclusion, this report shows that the selection of LAB enriched in genes populating pathways involved in indole metabolism can lead to the production of a yoghurt product that, when fed to pregnant mice, activates the intestinal immunity of their offspring. Although the composition of the yoghurt products and a functional cellular gene expression assay supported the role of indole derivatives and AhR activation in mediating these effects, an analysis of the metabolome of the products and murine samples indicated broader signaling mechanisms. Our findings thus raise interesting questions that should motivate future studies in animal models as well as in humans.”

References

45. Tan YQ, Wang YN, Feng HY et al. (2022) Host/microbiota interactions-derived tryptophan metabolites modulate oxidative stress and inflammation via aryl hydrocarbon receptor signaling. *Free Radic Biol Med* 184, 30-41.55.
- Sturino CF, Lachance N, Boyd M et al. (2006) Identification of an indole series of prostaglandin D2 receptor antagonists. *Bioorg Med Chem Lett* 16, 3043-3048.
46. Sturino CF, Lachance N, Boyd M et al. (2006) Identification of an indole series of prostaglandin D2 receptor antagonists. *Bioorg Med Chem Lett* 16, 3043-3048.
47. Ethridge AD, Bazzi MH, Lukacs NW, Huffnagle GB (2021) Interkingdom Communication and Regulation of Mucosal Immunity by the Microbiome. *J Infect Dis* 223, S236-s240.
48. Laudanno O, Vasconcelos L, Catalana J, Cesolari J (2006) Anti-inflammatory effect of bioflora probiotic administered orally or subcutaneously with live or dead bacteria. *Dig Dis Sci* 51, 2180-2183.

Note also the details of the methods to conduct the pathways analysis have been added on pages 26-27 of the section Material and Method as follows:

“Analysis of the pathways associated with the differentially expressed metabolic features was conducted with mummichog (tool available in MetaboAnalyst V6.0), a computational algorithms which predicts functional activity directly from spectral features without a priori identification of metabolites⁽⁶⁵⁾.”

Reference

65. Li S, Park Y, Duraisingham S et al. (2013) Predicting network activity from high throughput metabolomics. *PLoS Comput Biol* 9, e1003123.

Comment 3

Did the investigators measure any ILC3-related cytokines, like IL-22? It will be interesting to know whether the innate lymphoid cell population increase reflects the related cytokine production.

Response

Despite an excellent idea, we originally did not measure IL-22 in our experiments. Out of experience, it is not possible to see robust *Ii22* transcripts in bulk small intestinal RNA samples (Gomez de Agüero et al., Science, 2016), which we have now re-tested on ileal samples collected from the same mice used for flow cytometric analysis of intestinal immune cells. Again, no signal for *Ii22* was measurable in RT-qPCR. The other possibility of measuring intracellular IL-22 production in ILC3s using flow cytometry after an ex vivo stimulation is possible however very difficult with the cell numbers obtained on postnatal day 14. We analyzed this previously and saw that ILC3s produce the same amount of IL-22 on a per cell-level and that a reduction in total IL-22 producing cells is the consequence of lower abundance of ILC3s in the absence of AHR ligands in the milk (Gomez de Agüero et al., Science, 2016).

However, it was previously possible to see an increase in anti-microbial peptide expression in intestinal epithelial cells by RNA-Seq or RT-qPCR as an indirect measurement of IL-22 production that act on Paneth cells in the intestinal epithelium to induce the production of anti-microbial peptides. We have now performed RT-qPCR whole small intestinal tissue of these experiments to measure anti-microbial peptide expression as an indirect measurement of IL-22 production. As shown in Reviewer Figure 1 below, the expression level of several antimicrobial peptides was not different between the AhrY and ConY groups of pups. In addition, germ-free pups fed a plant-based chow instead of the purified diet containing ConY or AhrY showed clearly higher levels of AMP expression. One explanation for the absence of an increase in AMP expression in the presence of higher ILC3 levels may be that purified diet lack a co-factor that is required for AMP expression or intestinal epithelial cell development, present in chow, but not required for the increase in ILC3 cell numbers. Our previous reports are based on GF dams fed chow and supplemented with AhR ligands during pregnancy.

We feel that the addition of this information to the manuscript would significantly and unnecessarily lengthen the article (Methods, Results and Discussion) and have consequently not added this information in the revised manuscript. Instead, we now mention in the Discussion that we touch on the issue of downstream signaling of AhR with our analysis of the metabolome of the milk and serum of the mice as follows. See more details on the analysis of the metabolome in the response to comment 2 above.

Reviewer Figure 1: Antimicrobial peptide expression in ileal tissue of P14 pups born to ConY-diet or Ahr-diet fed dams.

Comment 4

What was the rationale of tissue harvesting on day 14 when mice were changed to a regular chow diet? It is unclear why the investigators did not harvest tissues/serum/milk on day 10 when mice were still on the AhrY diet.

Response

The experiment was designed according to previously published data where we found that AhR ligands originating from the maternal microbiota during pregnancy are transferred to the offspring via maternal milk and increase NKp46+ ILC3s in the small intestinal lamina propria (Gomez de Agüero et al., Science, 2016). Here, our goal was to mimic this phenotype by a nutritional alteration in the mother, namely via consumption of indole-rich yoghurt. As the earliest day showing a significance difference in NKp46+ ILC3 numbers in the small intestine of pups born to gestationally colonized versus germ-free dams was P14 (Figure 1C in Gomez de Agüero et al., Science, 2016), we also performed our analysis on postnatal day 14. Murine pups are nursed for a period of 21 days. However, they start eating solid food that drops to the bottom of the cage from about 14 days of age. In order to confine the yogurt consumption to the mother, we removed the yoghurt-containing diet on P10 and switched both groups back to regular chow. Milk and serum collection however, had to be done while the dams were still consuming the yoghurt-containing diet, i.e. on postnatal day 10.

This information was partly provided in the section “In vivo studies” of the Material and Methods but has now been revised to clearly explain why the tissue analysis was conducted on day 14 on pages 22-23 of the Materials and Methods as follows:

“As the earliest day showing a significance difference in NKp46+ ILC3 numbers in the small intestine of pups born to gestationally colonized versus germ-free dams was day 14 in the study of Gomez de Agüero and colleagues⁽³⁰⁾, we also performed the tissue analysis on postnatal day 14.

Also, page 10 in the Material and Methods now states that:

“...dams were fed a conventional chow diet, to prevent the pups from eating the yoghurt-containing diet themselves which is possible from postnatal day 12 onwards.

Reference

30. Gomez de Agüero M, Ganal-Vonarburg SC, Fuhrer T et al. (2016) The maternal microbiota drives early postnatal innate immune development. Science 351, 1296-1302.

Reviewer #2 (Comments for the Author)

In this paper, the LAB that can activate AhR were screened to ferment yogurt. The fermented product was added to the diet of pregnant dams to analyze its effect on the innate immune cells of offspring intestine. The article is well written. There are some minor questions.

Comment 1

The study lacks the ingredients of the animal diets. What is the amount of indole metabolites in the AhrY-diet?

Response

The composition of the diet as formulated by the manufacturer has now been added as a supplementary table to the manuscript (Table S2).

The analysis of the indole metabolites is not part of the standard composition measured by the manufacturer of the murine diet. We have now addressed this question on pages 9-10 of the Results section “Production and composition of the purified diets containing yoghurt for mice” as follows:

“Among the eight indole derivatives differing in the yoghurts, tryptophol and nicotinic or picolinic acid were significantly higher in the AhrY-diet compared to the ConY-diet (Fig. 2J-2K). Significant differences in tryptophan were not measured (Fig. 2I) and the remaining five metabolites could not be

detected in the murine diets. These findings confirmed the selection of the AhR- and control yoghurts for formulation of the murine diets for functional tests.”

These results were also addressed on pages 13-14 of the Discussion as follows:

“Among the seven metabolites enriched in the AhR yoghurt only nicotinic or picolinic acid, and tryptophol were measured as significantly enriched in the AhrY-diet compared to the ConY-diet. This discrepancy could be attributed to several factors. Firstly, the diet used for the incorporation of the yoghurts evidently already contains metabolites of the tryptophan pathway what might influence their relative distribution between the AhrY- and ConY-diets; secondly, the efficiency of the extraction of metabolites is strongly influenced by the matrix and this effect might be particularly important when comparing the composition of the dried pellets with the semi-liquid yoghurt; thirdly, as the composition of the yoghurts and murine diets were analyzed on samples stored at -20°C for different time periods, part of the differences in the content of indole derivatives between yoghurt and pellets might be due to different storage times; finally, although irradiation is a well-accepted method to sterilize diets for studies with specific pathogen-free animals, this method also induces changes in the nutritional composition of the diets⁽⁴¹⁾ what may have impacted on the concentration profile of the metabolites of the tryptophan pathway⁽⁴²⁾. However, taken together, our analytical results are consistent with an increase in metabolites of the tryptophan pathway in the yoghurt and diet prepared with the bacterial strains selected to increase the production of indole derivatives.”

References

41. Caulfield CD, Cassidy JP, Kelly JP (2008) Effects of gamma irradiation and pasteurization on the nutritive composition of commercially available animal diets. *J Am Assoc Lab Anim Sci* 47, 61-66.
42. Takano Y, Kaneko T, Kobayashi K et al. (2004) Experimental verification of photostability for free- and bound-amino acids exposed to γ -rays and UV irradiation. *Earth, Planets and Space* 56, 669-674.

Note also that the details of the methods used to extract (pages 24-25) and analyze (pages 25-26) the pellets have been added in different parts of the section “UHPLC-MS metabolomics analysis” of the Material and Method.

Comment 2

Dams are dietary intake of AhrY-diet containing indole metabolites, what are the concentrations of these metabolites in Dam's serum and breast milk? So the pathways that these metabolites affect the intestine innate immune could be analyzed.

Response

We have now addressed these comments as follows:

Revision of paragraph “The composition of milk and serum of germ-free dams fed the AhrY-diet differs from the one of dams fed the ConY-diet” on page 11 of the Results section:

“On postnatal day 10, serum and breast milk were collected and the samples were analyzed by mass spectrometry for indole derivatives and their untargeted metabolome. No significant differences in the relative concentration of the indole derivatives were observed between the experimental groups. However, the untargeted multivariate analysis of the metabolomes (Fig. 4 and Fig. S3) could clearly differentiate serum (A) and breast milk (B) of dams fed the AhrY-diet from dams fed the ConY-diet. The untargeted metabolome analysis could also differentiate the tested foods, i.e., the AhR yoghurt from the control yoghurt (C) as well as the AhrY-diet from the ConY-diet (D).

“Pathway analysis of mice milk and serum metabolomes using the mummichog tool revealed four pathways differentiating murine milk from AhrY-diet fed dams from control dams: linoleate metabolism (p value = 8.71×10^{-5}), prostaglandin formation from arachidonate (p value = 0.011), glycerophospholipid metabolism (p value = 0.023) and, fatty acid metabolism (p value = 0.038).”

In the Discussion section, we have also added the following paragraph on pages 15-16:

“Given the pleotropic activity of downstream signaling of AhR⁽⁴⁵⁾, identifying downstream effectors of activation of this receptor is a complex and technically difficult task. To undertake a first step in this direction we have complemented the analysis of the metabolites of the tryptophan pathway in the milk

and serum samples of the dams with an untargeted analysis of their metabolome. A multivariate analysis of the metabolomes revealed a significant difference between the test and control groups in both the milk and serum samples. Interestingly, a pathway analysis with mummichog identified 'prostaglandin formation from arachidonate' as significantly different between the milk metabolome of the mice fed the AhrY-diet compared to the group fed the ConY-diet. This result is interesting in light of the ability of AhR ligands to act as antagonists of the prostaglandin D2 receptor⁽⁴⁶⁾. On the other hand, prostaglandins also possess immunomodulatory activities that are independent of AhR ligands⁽⁴⁷⁾. In line with these findings, we cannot exclude that the genomic selection of bacteria to produce fermented food matrices with enhanced functionality in vivo may extend beyond the targeted endpoint(s) so that fermentation products others than indoles could independently or in synergy contribute to the observed changes in ILC3 numbers. In particular, the potential anti-inflammatory activity of dead bacteria⁽⁴⁸⁾, which were evidently not removed from the irradiated pellets, could also be immunologically active. Finally, we note that the mummichog tool used to identify the prostaglandin pathway is based on an enrichment analysis of molecular masses in pathway databases. Although this tool does not directly lead to the identification of metabolites, it nonetheless allows to pinpoint functional properties potentially modulated by the dietary treatment, which could be the target of future research."

As well as the following conclusion on page 18-19 of the Discussion:

"In conclusion, this report shows that the selection of LAB enriched in genes populating pathways involved in indole metabolism can lead to the production of a yoghurt product that, when fed to pregnant mice, activates the intestinal immunity of their offspring. Although the composition of the yoghurt products and a functional cellular gene expression assay supported the role of indole derivatives and AhR activation in mediating these effects, an analysis of the metabolome of the products and murine samples indicated broader signaling mechanisms. Our findings thus raise interesting questions that should motivate future studies in animal models as well as in humans."

References

45. Tan YQ, Wang YN, Feng HY et al. (2022) Host/microbiota interactions-derived tryptophan metabolites modulate oxidative stress and inflammation via aryl hydrocarbon receptor signaling. *Free Radic Biol Med* 184, 30-41.55.
- Sturino CF, Lachance N, Boyd M et al. (2006) Identification of an indole series of prostaglandin D2 receptor antagonists. *Bioorg Med Chem Lett* 16, 3043-3048.
46. Sturino CF, Lachance N, Boyd M et al. (2006) Identification of an indole series of prostaglandin D2 receptor antagonists. *Bioorg Med Chem Lett* 16, 3043-3048.
47. Ethridge AD, Bazzi MH, Lukacs NW, Huffnagle GB (2021) Interkingdom Communication and Regulation of Mucosal Immunity by the Microbiome. *J Infect Dis* 223, S236-s240.
48. Laudanno O, Vasconcelos L, Catalana J, Cesolari J (2006) Anti-inflammatory effect of bioflora probiotic administered orally or subcutaneously with live or dead bacteria. *Dig Dis Sci* 51, 2180-2183.

Note also the details of the methods to conduct the pathways analysis have been added on pages 26-27 of the section Material and Method as follows:

"Analysis of the pathways associated with the differentially expressed metabolic features was conducted with mummichog (tool available in MetaboAnalyst V6.0), a computational algorithms which predicts functional activity directly from spectral features without a priori identification of metabolites⁽⁶⁵⁾."

Reference

65. Li S, Park Y, Duraisingham S et al. (2013) Predicting network activity from high throughput metabolomics. *PLoS Comput Biol* 9, e1003123.

Comment 4

The biological significance of the AhrY-diet having an effect only on specific to ILC3s Instead of ILC2 and T cell needs to be discussed in the discussion.

Response

We now address this point on page 16 of the Discussion as follows:

“In accordance with published data and knowledge in the field of mucosal immunology^(24; 30), only offspring intestinal ILC3 expansion and maintenance but not ILC2 numbers or function were affected by feeding the AhrY-diet to pregnant and lactating dams. In contrast, both Treg and Th17 cells have been shown to be negatively affected by the absence of AhR(49). However, these T cell subsets are very scarce in early life⁽⁵⁰⁾ and hence our data conclusive with existing knowledge on AhR and intestinal immune cell development and homeostasis.”

References

24. Kiss EA, Vonarbourg C, Kopfmann S et al. (2011) Natural aryl hydrocarbon receptor ligands control organogenesis of intestinal lymphoid follicles. *Science* 334, 1561-1565.
30. Gomez de Agüero M, Ganai-Vonarburg SC, Fuhrer T et al. (2016) The maternal microbiota drives early postnatal innate immune development. *Science* 351, 1296-1302.
49. Quintana FJ, Basso AS, Iglesias AH et al. (2008) Control of T(reg) and T(H)17 cell differentiation by the aryl hydrocarbon receptor. *Nature* 453, 65-71.
50. Wagner C, Torow N, Hornef MW, Lelouard H (2022) Spatial and temporal key steps in early-life intestinal immune system development and education. *Febs j* 289, 4731-4757.

Additional changes not requested by the reviewers

Pages 1-2: New co-authors, and a new institution, have been added to the original list of co-authors (page 1) and author contributions (page 2) because of their contribution to the new experimental work conducted in response to the reviewer's comments.

Pages 3-4: A revised version of the section "Importance" was added to the abstract.

The legends of Figures 4 (page 37) and S3 (pages 38-39) have been adapted due to the panels added in response to the reviewer's comments.

A few minor changes (in red) were made to harmonize the wording across the manuscript (e.g. "pellet" was used to replace different wording used indiscriminately such as "murine pellet", "food pellet", "dietary pellet").

The formatting of the manuscript was adapted to make it fit better in the format requested by the editor (font size, double spacing...).

We have clarified in the section "UHPLC-MS metabolomics analysis" of the Material and Methods on page 24 that milk and serum were collected on postnatal day 10:

"Protein precipitation of murine serum (postnatal day 10), murine milk (postnatal day 10), and food products (milk, control yoghurt, AhR yoghurt and pellets) was obtained with the addition 1:4 (vol/vol) of acetonitrile containing 1% (vol/vol) formic acid."

Re: Spectrum00393-24R1 (Maternal consumption of yoghurt activating the aryl hydrocarbon receptor increases group 3 innate lymphoid cells in murine offspring.)

Dear Dr. Guy Vergeres:

Thank you for the privilege of reviewing your work. Below you will find my comments, instructions from the Spectrum editorial office, and the reviewer comments.

Authors have responded to the reviewers comments. Minor comments are provided from the new analyses. Also, with the next version submission update the Figure 2 as needed.

Thank you
Laxmi

Revision Guidelines

Sincerely,
Laxmi Yeruva
Editor
Microbiology Spectrum

Reviewer #1 (Comments for the Author):

Dear Authors,
One of the reviewers raised a concern, please see below.

"I did not see Figure 2J-2K that the authors had mentioned in the revised manuscript. These figures are important to address my comments from the first review". Can you please confirm this?

Please address this comment.

Thank you
Laxmi

I thank the authors for their efforts to address my comments on the previous version of the manuscript. The authors performed some new experiments and addressed my comments.

Minor comments:

- 1) Please reword the Following:

“The composition of milk and serum of germ-free dams fed the AhrY-diet differs from the one of dams fed the ConY-diet.” This title is not easy to understand.

What about changing to the following?

“The **composition** of the milk and serum of germ-free dams fed the AhrY-diet differs from that of dams fed the ConY-diet.”

- 2) Please include the figure or table number for the pathway analysis section (the section before the Discussion). The authors mentioned that some pathways differentiate murine milk from two different diets fed dams, but I do not know which figure or table they are referring to.
- 3) This sentence from the discussion section is not clear:

“The here presented experimental set-up was aligned with experiments performed previously,”

Thank you again.

Point-by-point response to the comments of reviewer

"Maternal consumption of yoghurt activating the aryl hydrocarbon receptor increases group 3 innate lymphoid cells in murine offspring."

Comment

We thank the authors for their efforts to address my comments on the previous version of the manuscript. The authors performed some new experiments and addressed my comments.

Response

Thank you very much for the considerate review, which has helped us to further improve our manuscript. Please find below our detailed responses to your comments.

Minor comments:

Comment

1) Please reword the Following:

"The composition of milk and serum of germ-free dams fed the AhrY-diet differs from the one of dams fed the ConY-diet." This title is not easy to understand. What about changing to the following? "The composition of the milk and serum of germ-free dams fed the AhrY-diet differs from that of dams fed the ConY-diet."

Response

We have made the change on page 11 of the manuscript.

Comment

2) Please include the figure or table number for the pathway analysis section (the section before the Discussion). The authors mentioned that some pathways differentiate murine milk from two different diets fed dams, but I do not know which figure or table they are referring to.

Response

We have now added the supplementary figure Fig. S4, referred to it in the results on page 11 of the manuscript, and added the legend of the figure in the file containing the legends of the supplementary material.

Comment

3) This sentence from the discussion section is not clear:

"The here presented experimental set-up was aligned with experiments performed previously,"

Response

We have now revised the sentence on page 14 as follows:

"The experimental set-up presented here was adapted from previous experiments, where germ-free pregnant mice were orally exposed to either a commensal *E. coli* strain producing indoles or to a pure AhR ligand during gestation⁽³⁰⁾."

Additional change

To address the request of the Spectrum staff we have removed the legends of the supplementary material from the main manuscript and moved it to the file "Legends Supplementary Material_Pimentel_20240924".

Best regards

Guy Vergères, in the name of the co-authors

Re: Spectrum00393-24R2 (Maternal consumption of yoghurt activating the aryl hydrocarbon receptor increases group 3 innate lymphoid cells in murine offspring.)

Dear Dr. Guy Vergeres:

Your manuscript has been accepted, and I am forwarding it to the ASM production staff for publication. Your paper will first be checked to make sure all elements meet the technical requirements. ASM staff will contact you if anything needs to be revised before copyediting and production can begin. Otherwise, you will be notified when your proofs are ready to be viewed.

Sincerely,
Laxmi Yeruva
Editor
Microbiology Spectrum